# Eosinophils Contribute to Oral Tolerance via Induction of RORγt-Positive Antigen-Presenting Cells and RORγt-Positive Regulatory T Cells

**DOI:** 10.3390/biom14010089

**Published:** 2024-01-10

**Authors:** Shunjiro Kurihara, Kotaro Suzuki, Masaya Yokota, Takashi Ito, Yuki Hayashi, Ryo Kikuchi, Takahiro Kageyama, Kazuyuki Meguro, Shigeru Tanaka, Arifumi Iwata, Yoshiyuki Goto, Akira Suto, Hiroshi Nakajima

**Affiliations:** 1Department of Allergy and Clinical Immunology, Graduate School of Medicine, Chiba University, 1-8-1 Inohana, Chiba 260-8670, Japan; s.kurihara84@chiba-u.jp (S.K.); sueitoh@chiba-u.jp (T.I.); rkikuchi@chiba-u.jp (R.K.); t.kageyama@chiba-u.jp (T.K.); kazuyuki.meguro@chiba-u.jp (K.M.); stanaka@chiba-u.jp (S.T.); aiwata@chiba-u.jp (A.I.); suaki@faculty.chiba-u.jp (A.S.); 2Synergy Institute for Futuristic Mucosal Vaccine Research and Development (cSIMVa), Chiba University, Chiba 260-8670, Japan; y-gotoh@chiba-u.jp; 3Division of Molecular Immunology, Medical Mycology Research Center, Chiba University, Chiba 260-8670, Japan; 4Division of Pandemic and Post-Disaster Infectious Diseases, Research Institute of Disaster Medicine, Chiba University, Chiba 260-8670, Japan; 5Division of Infectious Disease Vaccine R&D, Research Institute of Disaster Medicine, Chiba University, Chiba 260-8670, Japan

**Keywords:** oral tolerance, eosinophil, RORγt^+^ Tregs, RORγt^+^ APCs

## Abstract

Oral tolerance has been defined as the specific suppression of immune responses to an antigen by prior oral administration of the antigen. It has been thought to serve to suppress food allergy. Previous studies have shown that dendritic cells (DCs) and regulatory T cells (Tregs) are involved in the induction of oral tolerance. However, the detailed mechanisms of Treg induction in oral tolerance remain largely unknown. Eosinophils have been recognized as effector cells in allergic diseases, but in recent years, the diverse functions of tissue-resident eosinophils have been reported. Eosinophils in the intestine have been reported to induce Tregs by releasing TGF-β, but the role of eosinophils in oral tolerance is still controversial. In this study, we analyzed the roles of eosinophils in oral tolerance using eosinophil-deficient ΔdblGATA mice (mice lacking a high-affinity GATA-binding site in the GATA1 promoter). ΔdblGATA mice showed impaired antigen-induced oral tolerance compared to wild-type mice. The induction of RORγt^+^ Tregs in mesenteric lymph nodes (MLNs) by oral tolerance induction was impaired in ΔdblGATA mice compared to wild-type mice. An increase in RORγt^+^ antigen-presenting cells (APCs), which are involved in RORγt^+^ Treg differentiation, in the intestine and MLNs was not seen in ΔdblGATA mice. Notably, the expansion of group 3 innate lymphoid cells (ILC3s), a subset of RORγt^+^ APCs, by oral tolerance induction was seen in wild-type mice but not ΔdblGATA mice. These results suggest that eosinophils are crucial in the induction of oral tolerance, possibly via the induction of RORγt^+^ APCs and RORγt^+^ Tregs.

## 1. Introduction

Oral tolerance, defined as the specific suppression of the immune responses to an antigen by prior oral administration, has been thought to play a pivotal role in suppressing food allergy [1]. Classical dendritic cells (cDCs) and regulatory T cells (Tregs) in the mesenteric lymph nodes (MLNs) have been demonstrated to be involved in the induction of oral tolerance [2]. cDCs (MHCII^+^ CD11c^+^ CD64^−^ cells) in MLNs are classified as resident cDCs (MHC II^int^) or migratory cDCs (MHC II^high^), and the migratory cDCs, the most potent tolerogenic subset, present antigens to their cognate CD4^+^ T cells and convert these CD4^+^ T cells into Tregs by releasing TGF-β [3]. Recently, CD11b^−^ cDCs, a subset of MHC II^high^ migratory cDCs, have been shown to play a crucial role in the induction of peripherally induced Tregs (iTregs) during oral tolerance [4].

Recently, two major subsets of intestinal Tregs, namely, Helios^+^ NRP1^+^ subset and RORγ^+^ subset, have been characterized. Helios^+^ Tregs are abundant in the small intestine, while RORγ^+^ Tregs are abundant in the large intestine [5,6,7]. Helios^+^ Tregs differentiate in the thymus [8], and Helios^+^ GATA3^+^ Tregs, a part of Helios^+^ Tregs, are thought to be involved in tissue repair [9]. RORγ^+^ Tregs differentiate from peripheral conventional CD4^+^ T cells via Foxp3 induction, and unlike Helios^+^ Tregs, gut bacteria are robustly involved in their differentiation [7,10,11]. The microbiota-specific RORγ^+^ Tregs are thought to contribute to tolerance to commensal and pathogenic bacteria. Moreover, RORγ^+^ Tregs have been reported to have a prominent role in establishing oral tolerance to food allergy [12]. Recently, a new family of antigen-presenting cells (APCs), which are vital in the induction of intestinal RORγ^+^ Tregs, have been identified [13,14,15]. These cells express high levels of MHC II and RORγt and include a subset of group 3 innate lymphoid cells (ILC3s), extrathymic autoimmune regulator (AIRE)-expressing cells (eTACS), and DC-like cells [13,14,15]. However, the role of RORγt^+^ APCs in oral tolerance is still unknown.

Eosinophils have long been recognized as effector cells that cause tissue damage in type 2 immunity-driven diseases. Eosinophils are derived from bone marrow progenitor cells, and their differentiation requires IL-5 stimulation and high expression of GATA1 [16,17]. ΔdblGATA mice, in which the palindromic GATA-site in the GATA-1 promoter was disrupted, have complete ablation of the eosinophil lineage and have contributed significantly to examining eosinophils’ roles in vivo [18]. However, ΔdblGATA mice are not perfect tools and have been shown to exhibit some abnormalities other than eosinophils. For example, it has been shown that ΔdblGATA mice have mild anemia and mild impairment in basophil development but no apparent anomaly in mast cells [19,20]. In addition, using ΔdblGATA mice, conflicting results have been reported in several issues, such as intestinal mucus secretion [21,22,23], Peyer’s patch (PP) development [21,22,24], IgA production [21,22,24], and IgE production [21,24,25]. Despite these limitations, ΔdblGATA mice are nevertheless considered valuable tools for revealing the diverse functions of eosinophils.

In recent years, it has become clear that eosinophils reside in various tissues even under non-inflammatory conditions and have diverse tissue-specific functions [16,17]. In the gastrointestinal tract, eosinophils are present in 5–25% of the leukocyte fraction and have various functions [17]. Intestinal eosinophils interact with the mucosal immune system, epithelium, and microbiota to maintain homeostasis.

Regarding the action of intestinal eosinophils on immune responses, eosinophils have been shown to secrete the IL-1 receptor antagonist to inhibit IL-17 production by CD4^+^ T cells [26]. Eosinophils also promote the development of IgA-producing plasma cells in the small intestine by producing IL-6 and APRIL [24,27]. In addition, TGF-β derived from intestinal eosinophils is vital in the induction of microbiota-specific RORγ^+^ Tregs [28]. Several studies have also examined the contribution of eosinophils on oral tolerance [21,24]; however, the role of eosinophils in oral tolerance remains largely controversial, and the underlying mechanisms are poorly understood.

Here, we examine the roles of eosinophils in oral tolerance and assess the contribution of eosinophils on the differentiation of Tregs and RORγt^+^ APCs using eosinophil-deficient ΔdblGATA mice.

## 2. Materials and Methods

### 2.1. Experimental Animals

ΔdblGATA mice (mice deleted a high-affinity GATA-binding site in the GATA1 promoter), lacking eosinophils, on a BALB/c background were purchased from Jackson Laboratory (Bar Harbor, ME, USA) [18]. BALB/c mice (wild-type mice) were purchased from Charles River Laboratories (Atsugi, Kanagawa, Japan). The animal experiments in this study were approved by the Animal Committee of the Graduate School of Medicine, Chiba University (approval ID: A4-24). This study was carried out in compliance with ARRIVE guidelines. At the end of the experiments, mice were expertly euthanized by the cervical dislocation method for sample collection.

### 2.2. Induction of Oral Tolerance, Sensitization, and Food Allergy

The experimental procedure is shown in Figure 1A. To induce oral tolerance, mice (7–9-week-old) were freely given sterile distilled water containing 4 mg/mL of ovalbumin (OVA; chicken albumin, Grade V; Sigma-Aldrich (St. Louis, MO, USA)) as drinking water for 7 consecutive days (oral treatment (OT) group). Control animals were given sterile distilled water (control group). On days 14 and 28, mice were immunized intraperitoneally (i.p.) with 100 μg OVA absorbed in 1 mg of aluminum hydroxide gel (Thermo Fisher Scientific, Waltham, MA, USA). On days 33 and 35, mice were challenged by intragastric gavage (i.g.) with 40 mg OVA in 200 μL of PBS using tubing. Thirty minutes after the last OVA challenge on day 35, food allergy was assessed by a decrease in rectal temperature and a diarrhea score determined on the following scale: 0: normal (normal stool), 1: minimal (soft stool), 2: slight (slightly wet and soft stool), 3: moderate (wet and unformed stool with moderate perianal staining of the coat), and 4: severe (watery stool with severe perianal staining of the coat).

### 2.3. Serum IgE Levels

Sera were collected 2 h after the last challenge on day 35 and subjected to EIA for IgE measurement. Anti-OVA IgE was measured with an anti-OVA IgE EIA Kit (BioLegend, San Diego, CA, USA), and total IgE with a total IgE EIA Kit (FUJIFILM Wako Pure Chemical Corporation, Osaka, Japan).

### 2.4. Isolation of Cells from Small Intestinal Mucosa and Mesenteric Lymph Nodes

Two hours after the last challenge on day 35, mice were sacrificed, and fat tissue and Peyer’s patches were carefully removed from the proximal 1/3 of the small intestine. The small intestine was opened longitudinally, washed, and cut into pieces. The pieces were incubated in RPMI with 10% FCS and 1 mM EDTA at 37 °C to remove epithelial cells. The pieces were digested at 37 °C for 20 min in a shaking incubator with RPMI containing 10% FCS and 2 mg/mL collagenase (FUJIFILM Wako Pure Chemical Corporation, Doshomachi, Japan). The digested tissues were then passed through a 70 μm cell strainer, and cells in lamina propria (LP) and intraepithelial cells were collected from the interphase after 40/80% Percoll (Sigma-Aldrich) gradient centrifugation.

The mesenteric lymph nodes (MLNs) were collected and incubated in RPMI containing 10% FCS, 0.5 mg/mL collagenase, and 10 μg/mL DNase I (Roche, Basel, Switzerland). A single cell suspension was passed through a 70 μm cell strainer and washed with RPMI.

### 2.5. Flow Cytometry

Single cell suspensions obtained from the intestinal mucosa or MLNs were stained and analyzed on FACSCanto II (Becton Dickinson, Franklin Lakes, NJ, USA) using FlowJo software v10.8.2 (TreeStar, Ashland, OR, USA). The following antibodies were used: anti-B220 (RA3-6B2; BioLegend), anti-CD38 (90; BioLegend), anti-CD45 (30-F11; BioLegend), anti-GL7 (GL7; BioLegend), anti-CD138 (281-2; BioLegend), anti-CD4 (GK1.5; BioLegend), anti-CXCR5 (L138D7, BioLegend), anti-PD-1 (29F.1A12, BioLegend), anti-Siglec-F (E50-2440, BD Biosciences, Franklin Lakes, NJ, USA), anti-CD11c (N418, BioLegend), anti-I-A/I-E (M5/114.15.2, BioLegend), anti-IgE (RME-1, BioLegend), anti-CD103 (W19396D, BioLegend), anti-CD11b (M1/70, BioLegend), anti-IL-7Rα (A7R34, BioLegend), anti-CCR6 (29-2L17, BioLegend), anti-TCRβ (H57-597, BioLegend), and anti-TCRγδ (UC7-13D5, BioLegend). Dead cells were gated out using a Zombie NIR Fixable Viability Kit (BioLegend). Before staining, Fc receptors were blocked with an anti-CD16/32 antibody (2.4G2, BioLegend). Negative controls consisted of isotype-matched, directly conjugated, nonspecific antibodies (BD Biosciences). Intracellular staining was performed using anti-Foxp3 antibody (FJK-16s; Thermo Fisher Scientific) and anti-RORγt antibody (Q31-378; BD Biosciences), as described previously [29].

### 2.6. Histological Analysis

Intestinal tissue was fixed in 10% formalin, embedded in paraffin, and sectioned at 5 μm thickness for chloroacetate esterase (CAE) staining to detect mast cells, according to the standard protocol. CAE^+^ cells were quantified as described previously [30].

### 2.7. Microbiota Analysis

The contents of the small intestine were combined with Lysis Solution F (Nippon Gene, Toyama, Japan), crushed using a Shake Master Neo (BMS, Waseda, Japan), and incubated at 65 °C for 10 min. DNA was extracted from the solution using a Lab-Aid824s DNA Extraction kit (ZEESAN Biotech, Xiamen, China). Libraries were generated using a 2-step tailed PCR method, and the amplicons were sequenced on a MiSeq (Illumina, San Diego, CA, USA). The 16S rRNA reads were processed with QIIME2 (version: 2023.2). After trimming low-quality and chimeric reads, the feature-classifier plugin was utilized to compare the obtained representative sequences with the EzBioCloud 16S database for phylogenetic inference. Alignment and phylogeny plugins were then used to construct phylogenetic trees. Alpha and beta diversity analyses were conducted using the diversity plugin of QIIME2.

### 2.8. Statistical Analysis

Data are summarized as means ± SEMs. The statistical analysis of the results was performed using an unpaired *t*-test. Multiple group comparison was performed using 1-way or 2-way ANOVA or the Kruskal–Wallis test with GraphPad software v10 (GraphPad Software, La Jolla, CA, USA). *p*-values of less than 0.05 were considered significant.

### 2.9. Data Availability

The datasets generated for this study can be found in the DDBJ Sequence Read Archive (DRA017285).

## 3. Results

### 3.1. Eosinophils Are Involved in the Induction of Oral Tolerance to Food Allergy

To determine whether eosinophils participate in the induction of oral tolerance to food allergy, we first compared the consequence of oral tolerance induction in a murine model of food allergy in ΔdblGATA mice and wild-type mice [18]. Mice were orally administered OVA for 7 consecutive days to induce oral tolerance to OVA, and subsequently, these mice were sensitized by OVA intraperitoneally (i.p.) and challenged with OVA by intragastric gavage (i.g.) to induce food allergy (Figure 1A). Consistent with previous literature, oral administration of OVA prior to the sensitization improved the diarrhea score and attenuated rectal temperature reduction caused by i.g. administration of OVA in wild-type mice (Figure 1B,C). In contrast, oral administration of OVA prior to the sensitization did not improve the diarrhea score and rectal temperature reduction in ΔdblGATA mice (Figure 1B,C). In the absence of i.p. sensitization or i.g. administration of OVA, both wild-type mice and ΔdblGATA mice did not develop any symptoms. These results suggest that eosinophils are involved in the induction of oral tolerance to food allergy.

We also measured anti-OVA IgE levels in ΔdblGATA mice and wild-type mice seven days after final OVA sensitization. Oral administration of OVA prior to the sensitization reduced the levels of anti-OVA IgE and total IgE in wild-type mice but not in ΔdblGATA mice (Figure 2A,B). Mast cells are present in the gastrointestinal mucosa and are the primary effector cells of IgE-mediated allergic reactions to food [31]. Hence, we histologically analyzed the number of mast cells in the small intestinal mucosa seven days after the final OVA sensitization. As shown in Figure 2C, without oral administration of OVA prior to sensitization, there was no difference in the number of mast cells in the small intestinal mucosa between wild-type mice and ΔdblGATA mice. Under these conditions, the severity of food allergy (Figure 1) and anti-OVA IgE (Figure 2A) were comparable between wild-type mice and ΔdblGATA mice, suggesting that the mast cell degranulation capacity is not different between wild-type mice and ΔdblGATA mice. Moreover, with oral administration of OVA prior to sensitization, there was no difference in the number of mast cells in the small intestinal mucosa between wild-type and ΔdblGATA mice (Figure 2C). These results suggest that eosinophils are critical in the induction of oral tolerance, possibly via the reduction of antigen-specific IgE production.

### 3.2. Eosinophils Contribute to RORγt^+^ Treg Differentiation and Attenuate the Germinal Center Responses in Mesenteric Lymph Nodes

To clarify the mechanism underlying the impairment of OVA-specific IgE reduction by oral administration of OVA prior to the sensitization in ΔdblGATA mice (Figure 2A), we analyzed the cell population of MLNs seven days after the final OVA sensitization. T follicular helper (Tfh) cells and germinal center (GC) B cells in the germinal center are known to be critical in antigen-induced antibody production [32]. We, therefore, analyzed Tfh cells and GC B cells using flow cytometry. Oral administration of OVA prior to the sensitization reduced the number of Tfh cells (CXCR5^high^ PD-1^high^ CD4^+^ cells) and GC B cells (GL7^+^ CD38^−^ B220^+^ cells) in wild-type mice but not in ΔdblGATA mice (Figure 3A,B). Oral administration of OVA prior to the sensitization also reduced the number of total plasma cells (CD138^+^ CD38^+^ cells) in wild-type mice but not in ΔdblGATA mice (Figure 3C). Oral administration of OVA prior to the sensitization more significantly reduced IgE-producing plasma cells (CD138^+^ CD38^+^ IgE^+^ cells) in wild-type mice than in ΔdblGATA mice (Figure 3D). These results are consistent with the serum levels of IgE (Figure 2A,B).

CD103^+^ CD11b^−^ cDCs and Tregs in MLNs have been shown to participate in the induction of oral tolerance [4]. Therefore, we then analyzed CD103^+^ CD11b^−^ cDCs and Tregs. We found no significant differences in the number of CD103^+^ CD11b^−^ cDCs in the MLNs between wild-type and ΔdblGATA mice, with or without oral administration of OVA prior to the sensitization (Figure 3E). On the other hand, oral administration of OVA prior to the sensitization enhanced Treg differentiation in wild-type mice but not in ΔdblGATA mice (Figure 3F). We also examined RORγt^+^ Tregs, which are known to be necessary for the establishment of oral tolerance to food allergy [12]. Oral administration of OVA prior to the sensitization increased RORγt^+^ Tregs in wild-type mice but not in ΔdblGATA mice (Figure 3G). These results suggest that eosinophils contribute to RORγt^+^ Treg differentiation and suppress the differentiation of Tfh cells, GC B cells, and plasma cells in MLNs.

### 3.3. Eosinophils Are Crucial for the Differentiation of RORγt^+^ APCs

Next, we analyzed the cells in the small intestinal mucosa. We found that the number of eosinophils was not significantly changed by oral administration of OVA prior to the sensitization in wild-type mice (Figure 4A). We also confirmed the absence of eosinophils in ΔdblGATA mice even after oral administration of OVA prior to the sensitization (Figure 4A).

We then focused on RORγt^+^ APCs, a group of antigen-presenting cells recently identified in the gut that express high levels of RORγt and MHC II and induce RORγt^+^ Treg differentiation [13,14,15]. Oral administration of OVA prior to the sensitization increased the number of RORγt^+^ APCs, defined as TCRβ^−^ TCRγδ^−^ B220^−^ MHC II^high^ RORγt^+^ cells as described previously [13], in the small intestine in wild-type mice but not in ΔdblGATA mice (Figure 4B). Similar results were obtained for RORγt^+^ APCs in MLNs (Figure 4C). RORγt^+^ APCs include subsets of group 3 innate lymphoid cells (RORγt^+^ ILC3s), extrathymic autoimmune regulator (AIRE)-expressing cells (RORγt^+^ eTACS), and RORγt^+^ DC-like cells [13,14,15]. We then analyzed RORγt^+^ APC subsets in MLNs. Oral administration of OVA prior to the sensitization increased the number of RORγt^+^ ILC3s (IL-7Rα^+^ CCR6^+^ cells) but not RORγt^+^ DC-like cells (IL-7Rα^−^ CCR6^−^ cells) or RORγt^+^ eTACs (IL-7Rα^−^ CCR6^+^ cells) in wild-type mice, while oral administration of OVA prior to the sensitization did not increase any subsets in ΔdblGATA mice (Figure 4D). These results indicate that eosinophils promote RORγt^+^ ILC3s in oral tolerance induction.

### 3.4. The Microbiome in the Small Intestine Is Similar between Wild-Type and ΔdblGATA Mice

The gut microbiome in the intestine is crucial for the differentiation of RORγt^+^ APCs [13,14,15]. Since eosinophils are especially numerous in the small intestinal mucosa, we analyzed the microbiome in the small intestine in ΔdblGATA mice. We collected small intestinal contents from wild-type mice and ΔdblGATA mice and then performed 16s rRNA gut microbiota analysis (Figure 5A). There was no significant difference in microbiota composition of contents between wild-type mice and ΔdblGATA mice (Figure 5B,C). Moreover, no significant differences were found in the microbiota, with or without oral administration of OVA prior to the sensitization (Figure 5B,C). These results suggest that the microbiome was not significantly involved in the eosinophil-mediated increase of RORγt^+^ APCs in oral tolerance induction.

## 4. Discussion

This study demonstrates that the ΔdblGATA mice, which lack eosinophils, do not acquire oral tolerance to food allergy (Figure 1 and Figure 2), as observed in wild-type mice. We also found that the expansion of RORγt^+^ Tregs in MLNs by oral tolerance induction was impaired in ΔdblGATA mice (Figure 3). The differentiation of RORγt^+^ APCs in the small intestine and MLNs was also impaired in ΔdblGATA mice (Figure 4). On the other hand, no significant differences were observed in the microbiome in the small intestine between ΔdblGATA mice and wild-type mice (Figure 5). These results suggest that eosinophils play a pivotal role in the induction of oral tolerance, possibly via the expansion of RORγt^+^ APCs and RORγt^+^ Tregs.

We found that the expansion of RORγt^+^ Tregs in MLNs by oral tolerance induction was impaired in ΔdblGATA mice (Figure 3). Approximately 20–30% of Tregs in the intestinal tract express RORγt [33]. Epigenetic analyses have shown that Treg-specific epigenetic signature genes such as *Foxp3*, *Ctla-4*, *Gitr*, *Eos*, and *Helios* were significantly demethylated in RORγt^+^ Tregs [34]. These findings suggest that RORγt expression further stabilizes Treg lineage and functions. This notion is also supported by adoptive transfer experiments of RORγt^+^ Tregs in T cell transfer colitis models [34]. Recently, Fallegger et al. reported that eosinophil-derived TGF-β is vital for expanding RORγt^+^ Tregs in the gut [28]. They found that the expansion of RORγt^+^ Tregs in the gut upon bacterial exposure or antigen sensitization observed in control mice was impaired in inducible eosinophil-deficient mice and mice lacking eosinophil-specific TGF-β expression. Although they used a different strain of mice than the ΔdblGATA mice, their findings are consistent with our notion that eosinophils play a critical role in expanding RORγt^+^ Tregs by oral tolerance induction.

We also show that the increase of RORγt^+^ APCs in the small intestine and MLNs by oral tolerance induction partly depends on eosinophils (Figure 4). It has become clear that RORγt^+^ APCs have unique functions on CD4^+^ T cells that differ from those of conventional APCs [35]. Especially, RORγt^+^ APCs play a critical role in the induction of RORγt^+^ Tregs [13,14,15]. Recently, RORγt^+^ APCs are proposed to classify three subsets: RORγt^+^ ILC3s, RORγt^+^ eTACs, and RORγt^+^ DC-like cells [35]. In this study, we found that eosinophils are involved in the increase of RORγt^+^ ILC3s but not RORγt^+^ eTACs or RORγt^+^ DC-like cells (Figure 4D). These results suggest that in addition to the direct induction of RORγt^+^ Tregs by TGF-β production by eosinophils [28], eosinophils may indirectly induce RORγt^+^ Treg differentiation via expanding RORγt^+^ APCs, especially RORγt^+^ ILC3s.

However, caution is needed when interpreting the results because we used ΔdblGATA mice as eosinophil-deficient mice. For example, Hwang et al. found that ΔdblGATA mice were resistant to experimental autoimmune encephalomyelitis (EAE), a murine model of multiple sclerosis, but they reported that the impaired antigen-presenting cell function but not the deficiency of eosinophils resulted in the resistance to EAE due to GATA-1’s roles on monocyte lineages [36]. Therefore, as for the role of eosinophils in inducing RORγt^+^ APCs, the influence of GATA-1’s roles on RORγt^+^ APCs could not be ruled out. Reconstitution experiments of eosinophils will be necessary to complement these results.

It is assumed that the gut microbiome is vital for developing RORγt^+^ APCs [35]. We found no significant differences in the small intestinal microbiota in wild-type mice and ΔdblGATA mice before and after oral tolerance induction (Figure 5), while oral tolerance induction increased the number of RORγt^+^ ILC3s in wild-type mice but not in ΔdblGATA mice (Figure 4). These results suggest that eosinophils increase RORγt^+^ ILC3s, independent of the effect on the small intestinal microbiota. However, the number of mice we could analyze was small, which may have affected the results.

Regarding the relationship between eosinophils and intestinal microbiota, there are several reports investigating the microbiome in the large intestine or feces by the presence or absence of eosinophils [21,22,23]. Jung et al. have reported that segmented filamentous bacteria are increased, and Bacteroides are decreased in ΔdblGATA mice [21], but Chu et al. have reported opposite results [22]. The reasons for the discrepancy among the studies remain unknown. These different results may be due to the influence of environmental factors of the facility or the genetic background of the mice. Further studies are needed to elucidate the relationship between eosinophils and intestinal microbiota in oral tolerance induction.

It is well established that the small intestine and MLNs are essential sites for oral tolerance induction [37]. Tfh cells are a subset of CD4^+^ T cells that reside in the germinal center of lymph nodes and are essential to produce high-affinity antigen-specific IgE and subsequent IgE-mediated allergic reactions. In this study, we found that eosinophils are involved in the improvement of food allergy (Figure 1), the decreased production of antigen-specific IgE (Figure 2), and a decrease in Tfh cells (Figure 3A) by oral tolerance induction. Recently, Tfh13 cells have been identified as a subset of Tfh cells playing crucial roles in antigen-specific IgE production and anaphylaxis through the production of IL-4 and IL-13 [38]. These results suggest that the suppression of Tfh cells, especially Tfh13 cells, by eosinophils may attenuate antigen-specific IgE production and improve food allergy by oral tolerance induction. Since Tregs and MHC II^high^ ILC3s have been reported to suppress Tfh cells and antibody production by GC B cells in lymph nodes [39,40], the impaired expansion of RORγt^+^ Treg (Figure 3C) and RORγt^+^ ILC3 (Figure 4D) by the absence of eosinophils during oral tolerance induction may result in the impaired suppression of Tfh cells.

CD103^+^ CD11b^−^ cDCs have strong Treg induction capacity via TGF-β and retinoic acid and are thought to be involved in establishing oral tolerance [4]. In the present study, we found no significant difference in the number of CD103^+^ CD11b^−^ cDCs between wild-type and ΔdblGATA mice, with or without the oral tolerance induction (Figure 3E). Nevertheless, we could not estimate the function of CD103^+^ CD11b^−^ cDCs in the present study. Thus, further studies are necessary to determine whether the function of CD103^+^ CD11b^−^ cDCs is altered by the absence of eosinophils in the future.

## 5. Conclusions

In conclusion, our data indicate that eosinophils play a critical role in the induction of oral tolerance, possibly via the induction of RORγt^+^ APCs and RORγt^+^ Tregs. Clarifying the detailed mechanisms of the induction of RORγt^+^ APCs and RORγt^+^ Tregs by eosinophils may lead to the development of a new therapeutic option for food allergy.

## Figures and Tables

**Figure 1 biomolecules-14-00089-f001:**
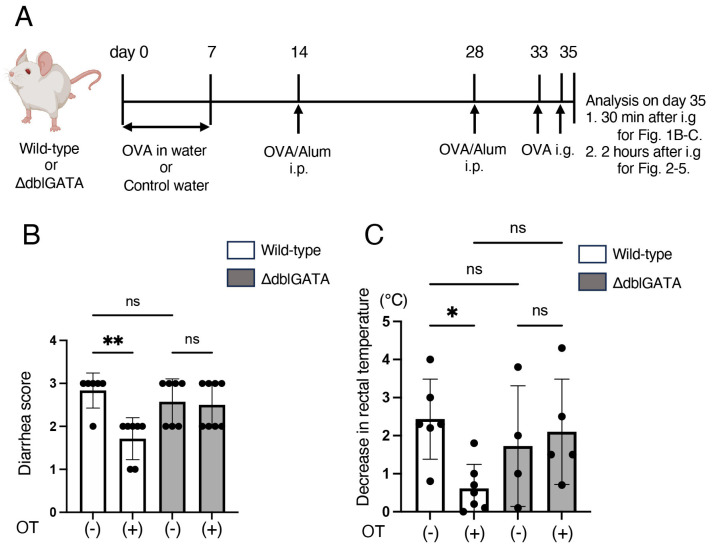
Eosinophils are involved in oral tolerance induction to food allergy. (**A**) Schematic drawing of the experimental protocol. ΔdblGATA mice and wild-type mice were freely given sterile water containing 4 mg/mL of OVA (OT(+)) or control sterile water (OT(-)) as drinking water for seven consecutive days. On days 14 and 28, mice were immunized intraperitoneally (i.p.) with OVA/alum. On days 33 and 35, mice were challenged by intragastric gavage (i.g.) with 40 mg OVA using tubing. (**B**,**C**) Thirty minutes after the last OVA challenge, food allergy was assessed by a diarrhea score (**B**) as described in Materials and Methods and a decrease in rectal temperature (**C**). Data are means ± SEM for 6–8 mice in each group. Data are compiled from three independent experiments. * *p* < 0.05, ** *p* < 0.01. ns: not significant. OT: oral treatment.

**Figure 2 biomolecules-14-00089-f002:**
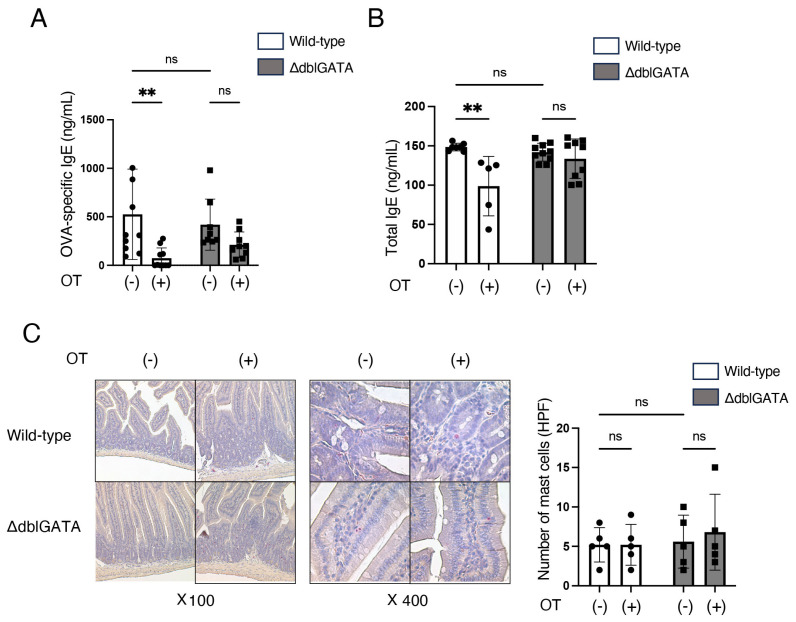
Eosinophils are crucial for the reduction of IgE by oral tolerance induction. Similar to Figure 1A, ΔdblGATA mice and wild-type mice were freely given sterile water containing OVA (OT(+)) or control sterile water (OT(-)) for seven consecutive days, immunized i.p. with OVA/alum on days 14 and 28, and challenged i.g. with OVA on days 33 and 35. Two hours after the last challenge on day 35, sera were collected and subjected to EIA for IgE measurement, and histological analyses were performed. Shown are the levels of anti-OVA-specific IgE (**A**) and total IgE (**B**). Data are means ± SEM for 5–10 mice in each group. Data are compiled from three independent experiments. ** *p* < 0.01. ns: not significant. (**C**) Representative histology depicting chloroacetate esterase-positive mast cells (**left**) and the numbers of chloroacetate esterase-positive mast cells in the small intestine (**right**). Data are means ± SEM for 5 mice in each group. Data are compiled from three independent experiments. ns: not significant.

**Figure 3 biomolecules-14-00089-f003:**
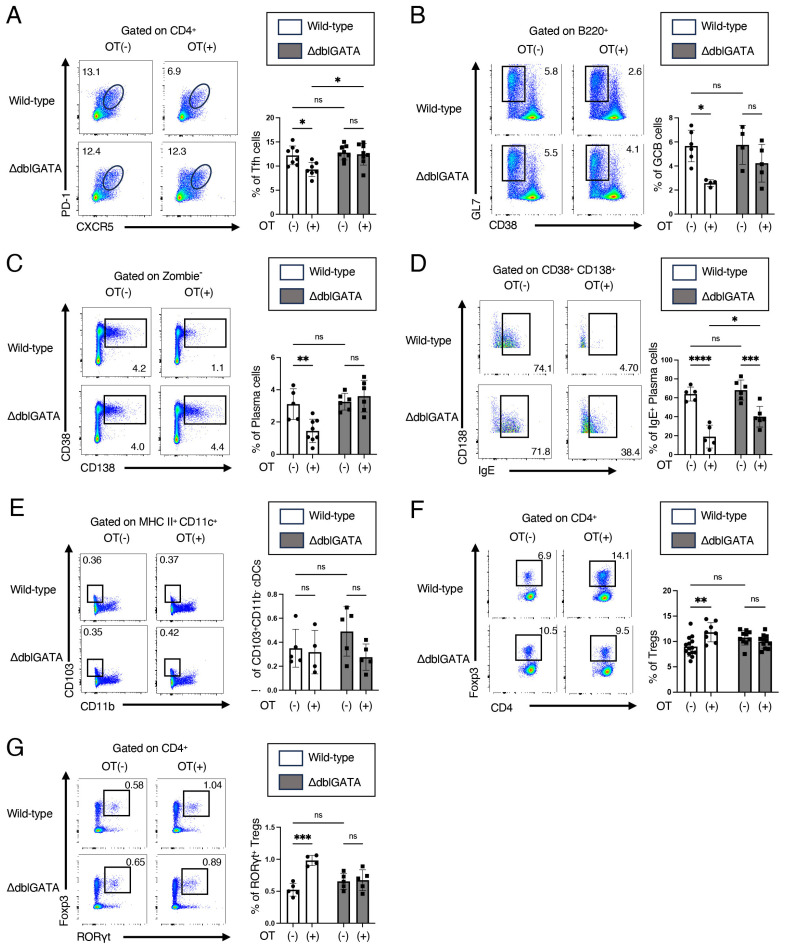
Eosinophils contribute to an increase in RORγt^+^ Tregs and attenuation of germinal center responses in mesenteric lymph nodes by oral tolerance induction. Similar to Figure 1A, ΔdblGATA mice and wild-type mice were freely given sterile water containing OVA (OT(+)) or control sterile water (OT(-)) for seven consecutive days, immunized i.p. with OVA/alum on days 14 and 28, and challenged i.g. with OVA on days 33 and 35. Two hours after the final challenge on day 35, mice were sacrificed, and cells from MLNs were analyzed using flow cytometry. (**A**) Representative dot plots of PD-1 vs. CXCR5 gated on CD4^+^ cells (**left**) and the frequencies of Tfh cells (CXCR5^high^ PD-1^high^ cells) in CD4^+^ cells (**right**). (**B**) Representative dot plots of GL7 vs. CD38 gated on B220^+^ cells (**left**) and the frequencies of GC B cells (GL7^+^ CD38^−^ cells) in B220^+^ cells (**right**). (**C**) Representative dot plots of CD38 vs. CD138 gated on live cells (**left**) and the frequencies of plasma cells (CD38^+^ CD138^+^ cells) in live cells (**right**). (**D**) Representative staining of CD138 vs. IgE gated on CD38^+^ CD138^+^ cells (**left**) and the frequencies of IgE^+^ plasma cells (IgE^+^ cells) in CD38^+^ CD138^+^ cells (**right**). (**E**) Representative dot plots of CD103 vs. CD11b gated on I-A/I-E (MHC II)^high^ CD11c^+^ cells (**left**) and the frequencies of CD11b^−^ cDCs (CD103^+^ CD11b^−^ cells) in live cells (**right**). (**F**) Representative dot plots of Foxp3 vs. CD4 gated on CD4^+^ cells (**left**) and the frequencies of Tregs (Foxp3^+^ CD4^+^ cells) in CD4^+^ cells (**right**). (**G**) Representative dot plots of Foxp3 vs. RORγt gated on CD4^+^ cells (**left**) and the frequencies of RORγt^+^ Tregs (Foxp3^+^ RORγt^+^ cells) in CD4^+^ cells (**right**). Data are compiled from three independent experiments, * *p* < 0.05, ** *p* < 0.01, *** *p* < 0.001, **** *p* < 0.0001. ns: not significant.

**Figure 4 biomolecules-14-00089-f004:**
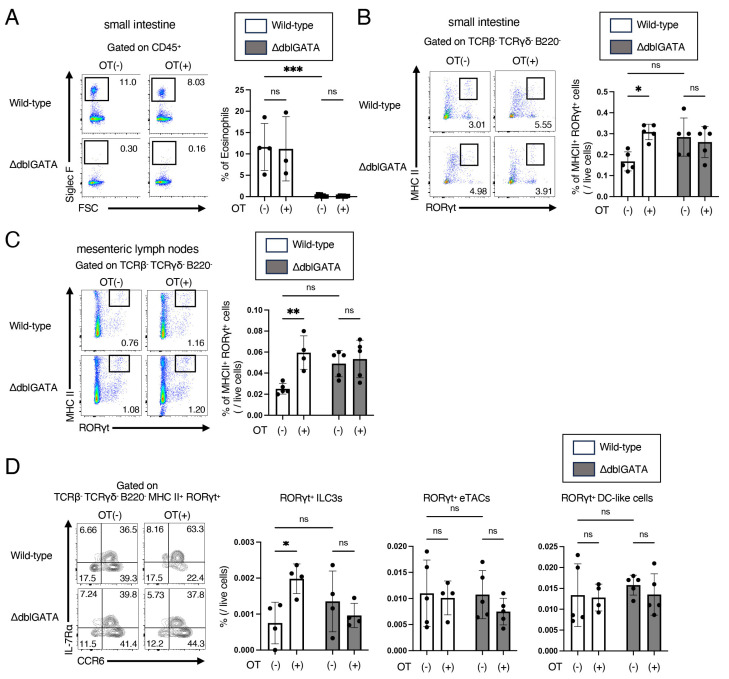
Eosinophils are crucial for the induction of RORγt^+^ APCs by oral tolerance induction. Similar to Figure 1A, ΔdblGATA mice and wild-type mice were freely given sterile water containing OVA (OT(+)) or control sterile water (OT(-)) for seven consecutive days, immunized i.p. with OVA/alum on days 14 and 28, and challenged i.g. with OVA on days 33 and 35. Two hours after the final challenge on day 35, cells were isolated from the proximal 1/3 of the small intestine and MLNs and analyzed using flow cytometry. (**A**) Representative dot plots of Siglec F vs. FSC gated on CD45^+^ cells (**left**) and the frequencies of eosinophils (Siglec F^+^ cells) in CD45^+^ cells (**right**) in the small intestine. (**B**,**C**) Representative dot plots of MHC II and RORγt gated on TCRβ^−^ TCRγδ^−^ B220^−^ cells (**left**) and the frequencies of RORγt^+^ APCs (TCRβ^−^ TCRγδ^−^ B220^−^ MHC II^high^ RORγt^+^ cells) in live cells (**right**) in the small intestine (**B**) and MLNs (**C**). (**D**) Representative staining of IL-7Rα vs. CCR6 gated on TCRβ^−^ TCRγδ^−^ B220^−^ MHC II ^high^ RORγt^+^ cells (**left**) and the frequencies of RORγt^+^ ILC3s (IL-7Rα^+^ CCR6^+^), RORγt^+^ eTACs (IL-7Rα^−/low^ CCR6^+^), and RORγt^+^ DC-like cells (IL-7Rα^−^ CCR6^−^) in live cells (**right**) in MLNs. Data are compiled from three independent experiments (n = 4–5), * *p* < 0.05, ** *p* < 0.01, *** *p* < 0.001. ns: not significant.

**Figure 5 biomolecules-14-00089-f005:**
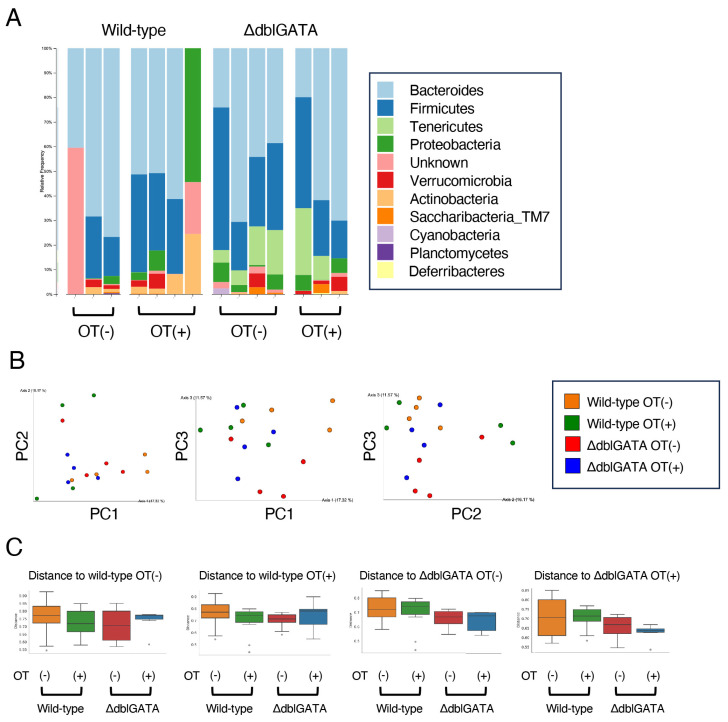
Eosinophils do not affect the microbiome in the small intestine. Similar to Figure 1A, ΔdblGATA mice and wild-type mice were freely given sterile water containing OVA (OT(+)) or control sterile water (OT(-)) for seven consecutive days, immunized i.p. with OVA/alum on days 14 and 28, and challenged i.g. with OVA on days 33 and 35. Two hours after the final challenge on day 35, DNA was extracted from the small intestine content, and 16S rRNA sequencing was performed. (**A**) Phylum-level relative abundance of small-intestine microbiota (n = 3–4). (**B**) Principal coordinate analysis plot of weighted UniFrac distances for small intestine microbiota from ΔdblGATA mice and wild-type mice (n = 3–4). PC1, PC2, PC3, principal coordinates 1, 2, and 3. (**C**) Bar plot of the difference in Beta-diversity distance for small intestine microbiota from ΔdblGATA mice and wild-type mice (n = 3–4).

## Data Availability

All data generated or analyzed during this study are included in this published article.

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
