# Peer review of "Eosinophils Contribute to Oral Tolerance via Induction of RORγt-Positive Antigen-Presenting Cells and RORγt-Positive Regulatory T Cells"

_biomolecules, 2024, doi:10.3390/biom14010089_

Round 1
Reviewer 1 Report
Comments and Suggestions for Authors
Nakajima's group demonstrated the potential pathway of oral tolerance contributed by Eosinophils in an eosinophils knockout mice model. This study is very interesting and important, which explored the oral tolerance of food allergy from a novel aspect and could be used as a new therapeutic option for food allergy.
A few questions remain..
Line 113 MLN collection and prep was not mentioned in the method.
Line 116 Epithelial cells were removed, however, IEL cells remained. The cells were the mixture of LP and IEL at the end of the collection.
Line 123 source of the cells is not clear.
Fig1A The end timepoint or the collection time was not clear. Line 202 “analyzed the number of mast cells in the small intestinal mucosa 7 days after the final OVA sensitization”, so assumed that the collection was performed on the last challenge day?
Fig1B, line 172, Diarrhea score are discrete data, need Kruskal-Wallis test.
Fig2A, B, line 197, IgE levels were measured at 7 days after final OVA sensitization, however, at day 33, the mice received the 1st challenge. What are the effects like after the first challenge? Is it possible that IgE levels, mast cells and other cells levels were disrupted after the first challenge but to receive another challenge in two days is not long enough to elicit another response?
Fig2D, line 202, to address the severity of IgE-mediated food allergy in this model, should use the degranulation of the mast cells rather than the number of mast cells.
Fig2,3 and 4 symbols of significance were not labeled.
Fig4B, Line 255, “Oral administration of OVA prior to the sensitization increased the number of RORγt+ APCs, defined as lineage-negative, MHC IIhigh, and RORγt+ cells” The definition of lineage-negative was not clear.
Basal levels of IgE, mast cells, cells detected by flow and microbiome in the ΔdblGATA mice were not provided. Does knockout of eosinophils impair the homeostasis and tolerance in mice before sensitization and tolerance build-up?
Comments on the Quality of English LanguageGood. minor editing required
Reviewer 2 Report
Comments and Suggestions for Authors
The paper presents very interesting results with a high degree of novelty. It would seem very suitable for a main stream immunology or allergy journal and indeed the subject is more about cellular immunology than biomolecules. This aside the conclusions are mostly consistent with the data and the experiments appear well executed with the correct techniques.
As a minor but important point the methods need to state whether the feeding was by gavage or by administration to the mouth.
The results with the CD103+ DCs need to be better explained in the text in both the results and discussion sections. Do they mean that the CD103+ DCs were not involved in the oral tolerance as described by others and the introduction of this paper?
The microbiota results do appear to show differences especially the appearance of Deferribacteres in the mutant and possibly increases Firmicutes due to the OA feeding in wild type mice. The number of mice seem underpowered. There are also outliers who have an obvious disruption to their microbiota with an unknown bacterium. Singh et al. (doi: 10.1111/imm.13110) have described differences in the microbiota of the dblGATA mutants so the differences with this paper need to be discussed. It is recommended that this study examines more mice for their microbiota.
A brief description of the phenotype of delta-dblGATA mice in the introduction would help, noting for the readers, that the except for the eosinophils the mutant mice, at homeostasis, have normal numbers of other haemopoietic lineages (which might not be expected given the key function of GATA-1).
Hwang et al. (doi: 10.1182/bloodadvances.2022008234.) describe that delta-dblGATA mice are resistant to experimental autoimmune encephalomyelitis but that it is not due to the lack of eosinophils. Firstly given the large number of covert effects that could be caused by GATA-1 depletion this should be discussed. Secondly the type of reconstitution experiments used in this study would be needed in further studies.
Round 2
Reviewer 1 Report
Comments and Suggestions for Authors
Thank you for replying my previous questions. I don't have any other concern. I am good with the present version.